# When in Doubt: Neural Non-Parametric Uncertainty Quantification for Epidemic Forecasting

**Harshavardhan Kamarthi**    **Lingkai Kong**    **Alexander Rodríguez**
**Chao Zhang**    **B. Aditya Prakash**
College of Computing
Georgia Institute of Technology
{harsha.pk,lkkong,arodriguezc,chaozhang,badityap}@gatech.edu

## Abstract

Accurate and trustworthy epidemic forecasting is an important problem for public health planning and disease mitigation. Most existing epidemic forecasting models disregard uncertainty quantification, resulting in mis-calibrated predictions. Recent works in deep neural models for uncertainty-aware time-series forecasting also have several limitations; *e.g.*, it is difficult to specify proper priors in Bayesian NNs, while methods like deep ensembling can be computationally expensive. In this paper, we propose to use neural functional processes to fill this gap. We model epidemic time-series with a probabilistic generative process and propose a functional neural process model called EPIFNP, which directly models the probability distribution of the forecast value in a non-parametric way. In EPIFNP, we use a dynamic stochastic correlation graph to model the correlations between sequences, and design different stochastic latent variables to capture functional uncertainty from different perspectives. Our experiments in a real-time flu forecasting setting show that EPIFNP significantly outperforms state-of-the-art models in both accuracy and calibration metrics, up to *2.5x* in accuracy and *2.4x* in calibration. Additionally, as EPIFNP learns the relations between the current season and similar patterns of historical seasons, it enables interpretable forecasts. Beyond epidemic forecasting, EPIFNP can be of independent interest for advancing uncertainty quantification in deep sequential models for predictive analytics.

## 1   Introduction

Infectious diseases like seasonal influenza and COVID-19 are major global health issues, affecting millions of people [14, 34]. Forecasting disease time-series (such as infected cases) at various temporal and spatial resolutions is a non-trivial and important task [34]. Estimating various indicators e.g. future incidence, peak time/intensity and onset, gives policy makers valuable lead time to plan interventions and optimize supply chain decisions, as evidenced by various Centers for Disease Control (CDC) prediction initiatives for diseases like dengue, influenza and COVID-19 [33, 16, 30].

Statistical approaches [5] for the forecasting problem are fairly new compared to more traditional mechanistic approaches [13, 38]. While valuable for 'what-if' scenario generation, mechanistic models have several issues in real-time forecasting. For example, they cannot easily leverage data from multiple indicators or predict composite signals. In contrast, deep learning approaches in this context are a novel direction and have become increasingly promising, as they can ingest numerous data signals without laborious feature engineering [37, 33, 1, 8].

However, there are several challenges in designing such methods, primarily with the need to handle uncertainty to give more reliable forecasts [14]. Decision makers need to understand the inherent uncertainty in the forecasts so that they can make robust decisions [32]. Providing probabilistic

forecasts and interpreting what signals cause the model uncertain is also helpful to better communicate the situation to the public. Due to the inherent complexity of the prediction problem, just like weather forecasting, so-called 'point' forecasts without uncertainty are increasingly seen as not very useful for planning for such high-stake decisions [14, 33].

Uncertainty quantification in purely statistical epidemic forecasting models is a little explored area. Most traditional methods optimize for accuracy of 'point-estimates' only. Some approaches that model the underlying generative distribution of the data naturally provide a probability distribution of the outputs [4, 5, 44, 32], but they do not focus on producing *calibrated* distributions [12, 22] as well. Another line of research addresses this problem with the use of simple methods such as an ensemble of models to build a sample of forecasts/uncertainty bounds [34, 6]. Recent attempts for deep learning forecasting models use ad-hoc methods such as bootstrap sampling [37]; while others disregard this aspect [42, 36]. As a result these can produce wildly wrong predictions (especially in novel/atypical scenarios) and can be even confident in their mistakes. In time-series analysis, while a large number of deep learning models [1] have been proposed, little work has been done to quantify uncertainty in their predictions. Bayesian deep learning [28, 3, 27] (and approximation methods [10, 25, 43]) and deep ensembling [24] are two directions that may mitigate this issue, but their applicability and effectiveness are still largely limited by factors such as intractable exact model inference [3, 27], difficulty of specifying proper parameter priors [26], and uncertainty underestimation [21, 19]. Neural Process (NP) [11] and Functional Neural Process (FNP) [26] are recent frameworks developed to incorporate stochastic processes with DNNs, but only for static data.

Our work aims to close these *crucial* gaps from both viewpoints. We propose a non-parametric model for epi-forecasting by 'marrying' deep sequential models with recent development of neural stochastic processes. Our model, called EPIFNP, leverages the expressive power of deep sequential models, while quantifying uncertainty for epidemic forecasting directly in the functional space. We extend the idea of learning dependencies between data points [26] to sequential data, and introduce additional latent representations for both local and global views of input sequences to improve model calibration. We also find that the dependencies learned by EPIFNP enable reliable interpretation of the model's forecasts.

Figure 1 shows an example of a well-calibrated forecast due to EPIFNP in flu forecasting. CDC is interested in forecasting weighted Influenza-like-illness (wILI) counts, where ILI is defined as "fever and a cough and/or a sore throat without a known cause other than flu. Figure 1 (a) shows the historical ILI data with abnormal seasons highlighted; Figure (b) shows how our method EPIFNP, in contrast to others, is able to react well to a particularly novel event (in this case, introduction of a symptomatically simi-lar COVID-19 disease), giving both *accurate* and *well-calibrated* forecasts.

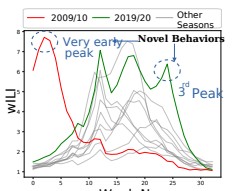

(a) Historical wILI seasons sequences, 2003-20

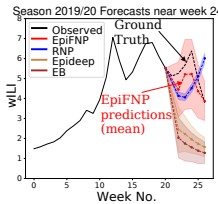

(b) Probabilistic predictions of all methods

Figure 1: EPIFNP *(red) is the only model reacting reliably for the atypical 3rd peak of 2019/20 season and whose 95% confidence bounds completely encloses the ground truth.*

Our main contributions are:
• **Probabilistic Deep Generative Model:** We design a neural Gaussian processes model for epidemic forecasting, which automatically learns stochastic correlations between query sequences and historical data sequences for non-parametic uncertainty quantification.
• **Calibration and Explainability:** EPIFNP models the output forecast distribution based on similarity between the current season and the historical seasons in a latent space. We introduce additional latent variables to capture global information of historical seasons and local views of sequences, and show that this leads to better-calibrated forecasts. Further, the relations learned between the current season and similar patterns from previous seasons enable explaining the predictions of EPIFNP.
• **Empirical analysis of accurate well-calibrated forecasting:** We perform rigorous benchmarking on flu forecasting and show that EPIFNP significantly outperforms strong baselines, providing up to *2.5x* more accurate and *2.4x* better calibrated forecasts. We also use outlier seasons to show the uncertainty in EPIFNP makes it adapt well to unseen patterns compared with baselines.

## 2 Problem and Background

We focus on epidemic disease forecasting in this paper. Our goal is to predict the disease incidence few week into the future given the disease surveillance dataset containing incidence from the past seasons as well as for the past weeks of the current season. This is formulated as a supervised time-series forecasting problem as follows.

**Epidemic Forecasting task:** Let the incidence for season $i$ at week $t$ be $x_i^{(t)}$. During the current season $N + 1$ and current week $t$, we first have the snippet of time-series values upto week $t$ denoted by $\mathbf{x}_{N+1}^{(1...t)} = \{x_{N+1}^{(1)}, \ldots, x_{N+1}^{(t)}\}$. We are also provided with data from *past* historical seasons 1 to $N$ denoted by $H = \{\mathbf{x}_i^{(1...T)}\}_{i=1}^N$ where $T$ is number of weeks per season. In *real-time* forecasting, intuitively our goal is to use all the currently available data, and predict the next few future values (usually till 4 weeks in future). That is to predict the value $y_{N+1}^{(t)} = x_{N+1}^{(t+k)}$, $k$ week in future where $k \in \{1, 2, 3, 4\}$ given $\mathbf{x}_{N+1}^{(1...t)}$ and $H$. Formally, our task is: *given (a) the dataset of historical incidence sequences $H$ and (b) snippet of incidence for current season $N + 1$ till week $t$, $x_{N+1}^{(1...t)}$, estimate an accurate prediction for $y_{N+1}^{(t)}$ and a well-calibrated probability distribution* $\hat{p}(y_{N+1}^{(t)} | \mathbf{x}_{N+1}^{(1...t)}, H)$. There are several ways to evaluate such forecasts [40], which we elaborate later in our experiments.

## 3 Our Methodology

**Overview:** EPIFNP aims to produce calibrated forecasting probabilistic distribution. One popular choice is to use BNNs [3, 9] which impose probability distributions for weight parameters. However, as Deep Sequential Models (DSMs) have an enormous number of uninterpretable parameters, it is impractical to specify proper prior distributions in the parameter space. Existing works usually adopt simple distributions [3, 35], e.g., independent Gaussian distribution, which could severely under-estimate the true uncertainty [21]. To solve this issue, we propose EpiFNP, which combines (1) the power of DSMs in representation learning and capturing temporal correlations; and (2) the power of Gaussian processes (GPs) in non-parametric uncertainty estimation directly in the functional space similar to [26], instead of learning probability distributions for model parameters.

During *training phase* of our supervised learning task, EPIFNP is trained to predict $x_i^{(t+k)}$ given $x_i^{(1...t)}$ as input for $i \leq N$. Therefore, we define the training set $M$ as set of partial sequences and their forecast ground truths from historical data $H$, i.e, $M = \{(\mathbf{x}_i^{(1...t)}, y_i^{(t)}) : i \leq N, t + k \leq T, y_i^{(t)} = x_i^{(t+k)}\}$. For simplicity, let $\mathbf{X}_M$ be set of the partial sequences in $M$ and $\mathbf{y}_M$ the set of ground truth labels. Following GPs for non-parametric uncertainty quantification, EPIFNP constructs the forecasting distribution on the historical sequences. Since the number of possible sequences that can be extracted from $H$ is prohibitively large, we narrow down the set of candidates into a set of sequences that comprehensively represents $H$, called the reference set $R$. We choose the set of full sequences of $T$ incidence values for each season as reference set, i.e, $R = \{\mathbf{x}_i^{(1...T)}\}_{i=1}^{N_R}$. We refer elements of $M$ as $\{\mathbf{x}_i^M, y_i^M\}_{i=1}^{N_M}$ and $R$ as $\{\mathbf{x}_i^M\}_{i=1}^{N_R}$ when we don't need to specify the week and season. Also let $\mathbf{X}_{\mathcal{D}} = \{\mathbf{x}_i^M\}_{i=1}^{N_M} \cup \{\mathbf{x}_i^R\}_{i=1}^{N_R}$, the union of reference and training sequences.

The generative process of EPIFNP includes three key steps (also see Figure 2 and Eq. 1):

(a) **Probabilistic neural sequence encoding** (Section 4.2). The first step of the generative process is to use a DSM to encode the sequence $\mathbf{x}_i \in \mathbf{X}_{\mathcal{D}}$ into a *variational* latent embedding $\mathbf{u}_i \in \mathbf{U}_{\mathcal{D}}$. The representation power of DSM helps us to model complex temporal patterns within sequences, while the probabilistic encoding framework enables us to capture the uncertainty in sequence embedding.

(b) **Stochastic correlation graph construction** (Section 4.3). The second step is to capture the correlations between reference ($\mathbf{U}_R$) and training ($\mathbf{U}_M$) data points in the *latent embedding space* (i.e. seasonal similarity between epidemic curves). We use a stochastic data correlation graph $\mathbf{G}$, which plays a similar role to the covariance matrix in classic GPs. It encodes the dependencies between reference and training sequences, enabling non-parametric uncertainty estimation.

(c) **Final predictive distribution parameterization** (Section 4.4). Finally, we parameterize the predictive distribution with three stochastic latent variables: (1) The global stochastic latent variable $\mathbf{v}$, which is shared by all the sequences. This variable captures the overall information of the

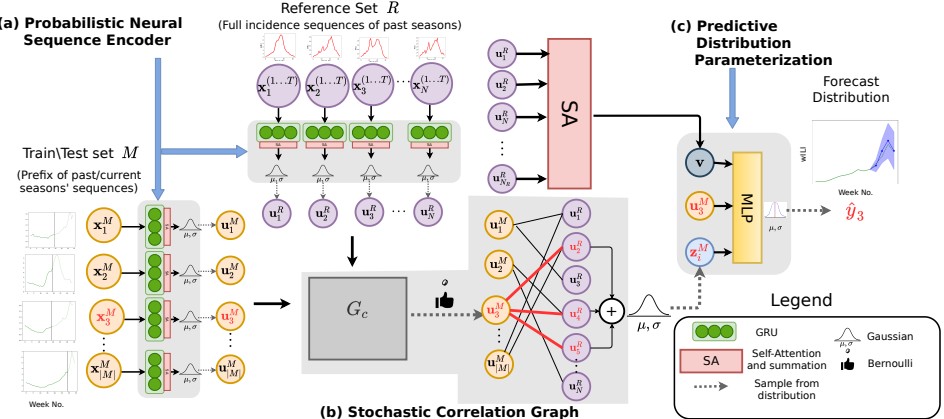

Figure 2: *Pipeline of proposed* EPIFNP *model. (i) Three main components (a), (b) and (c) correspond to the terms in Equation 1. (ii) Variables highlighted in* Red *correspond to steps specific to inference of sequence* $\mathbf{x}_3^M$.

underlying function based on all the reference points. (2) The local stochastic latent variables $\mathbf{Z}_M = \{\mathbf{z}_i^M\}_{i=1}^{N_M}$. This term captures the data correlation uncertainty based on the stochastic data correlation graph $\mathbf{G}$. (3) The stochastic sequence embeddings $\mathbf{U}_M = \{\mathbf{u}_i^M\}_{i=1}^{N_M}$. This term captures the embedding uncertainty and provide additional information beyond the reference set.

Hence, putting it all together from the generative process, we factorize the predictive distribution of the training sequences into three corresponding parts ($\theta$ is the union of the parameters in EPIFNP):

$$p(\mathbf{y}_M|\mathbf{X}_M, R) = \sum_{\mathbf{G}} \int \underbrace{p_\theta(\mathbf{U}_\mathcal{D}|\mathbf{X}_\mathcal{D})}_{(a)} \underbrace{p(\mathbf{G}|\mathbf{U}_\mathcal{D})}_{(b)}$$
$$\underbrace{p_\theta(\mathbf{Z}_M, |\mathbf{G}, \mathbf{U}_R)p_\theta(\mathbf{v}|\mathbf{U}_R)p_\theta(\mathbf{y}_M|\mathbf{U}_M, \mathbf{Z}_M, \mathbf{v})}_{(c)} \, d\mathbf{U}_\mathcal{D}d\mathbf{Z}_Md\mathbf{v}. \tag{1}$$

Compared to existing recurrent neural process (RNP) [31] for sequential data (and its related predecessors [11, 17]), our EPIFNP process has stronger representation power and is more interpretable. Specifically, RNP uses a single global stochastic latent variable to capture the functional uncertainty, which is not flexible enough to represent a complicated underlying distribution. In contrast, EPIFNP constructs three stochastic latent variables to capture the uncertainty from different perspectives and can interpret the prediction based on the correlated reference sequences.

### 3.1 Probabilistic Neural Sequence Encoder

The probabilistic neural sequence encoder $p_\theta(\mathbf{U}_D|\mathbf{X}_D)$ aims to model the complex temporal correlations of the sequence for accurate predictions of $y$, while capturing the uncertainty in the sequence embedding process. To this end, we design the *sequence encoder* as a RNN and stack *a self-attention layer* to capture long-term correlations. Moreover, following Variational auto-encoder (VAE) [18], we model the latent embedding $\mathbf{u}_i$ as a Gaussian random variable to capture embedding uncertainty.

We encode all the sequences, including reference sequences and training sequences, independently. Taking one sequence $\mathbf{x}_i$ as an example, we first feed $\mathbf{x}_i$ into a Gated Recurrent Unit (GRU) [7]:

$$\{\mathbf{h}_i^{(1)} \dots, \mathbf{h}_i^{(t)}\} = \text{GRU}(\{x_i^{(1)} \dots, x_i^{(t)}\}). \tag{2}$$

where $\mathbf{h}_i^{(t)}$ denotes the hidden state at time step $t$. To obtain the embedding of $\mathbf{x}_i$, the simplest way is to directly use the last step hidden state, $\mathbf{h}^{(t)}$. However, using the last step embedding is inadequate for epidemic forecasting as the estimates for ILI surveillance data are often delayed and revised multiple times before they stabilize [1]. Over-reliance over the last step hidden state would harm the predictive ability of the model. Therefore, we choose to use a self-attention layer [41] to aggregate the information of the hidden states across all the time steps:

$$\{\alpha_i^{(1)} \dots, \alpha_i^{(t)}\} = \text{Self-Atten}(\{\mathbf{h}_i^{(1)} \dots, \mathbf{h}_i^{(t)}\}), \qquad \bar{\mathbf{h}}_i = \sum_{t'=1}^{t} \alpha_i^{(t')}\mathbf{h}_i^{(t')}, \tag{3}$$

where $\bar{\mathbf{h}}_i$ is the summarized hidden state vector. Compared with the vanilla attention mechanism [2], self-attention is better at capturing long-term temporal correlations [41]. Though $\bar{\mathbf{h}}_i$ has encoded the temporal correlations, it is deterministic and cannot represent embedding uncertainty. Inspired by VAE, we parameterize each dimension of the latent embedding $\mathbf{u}_i$ as a Gaussian random variable:

$$p_\theta([\mathbf{u}_i]_k|\mathbf{x}_i) = \mathcal{N}([g_1(\bar{\mathbf{h}}_i)]_k, \exp([g_2(\bar{\mathbf{h}}_i)]_k)), \tag{4}$$

where $g_1$ and $g_2$ are two multi-layer perceptrons (MLPs), $[\cdot]_k$ is the $k$-th dimension of the variable.

## 3.2 Stochastic Data Correlation Graph

The stochastic graph $\mathbf{G}$ is used to model the correlations among sequences, which is central to the non-parametric uncertainty estimation ability of EPIFNP. It is realized by constructing a bipartite graph from the reference set $R$ to the training set $M$ *based on the similarity between their sequence embeddings*. With this graph, we aim to model the dynamic similarity among epidemic curves as in [1] but in a stochastic manner, which allows us to further quantify the uncertainty coming from our latent representations of the sequences. Note that the similarity with reference sequence embeddings dynamically changes across the current season since different periods of the season may be similar to different sets of reference sequences (as we illustrate in Section 4.4).

We first construct a complete weighted bipartite graph $\mathbf{G}_c$ from $R$ to $M$, where the nodes are the sequences. *The weight of each edge is calculated as similarity between two sequences in the embedding space using the radial basis function kernel* $\kappa(\mathbf{u}_i^R, \mathbf{u}_j^M) = \exp(-\gamma||\mathbf{u}_i^R - \mathbf{u}_j^M||^2)$. Modeling such a similarity in the embedding space is more accurate than in the input space by leveraging the representation power of the neural sequence encoder.

Though we can directly use $\mathbf{G}_c$ to encode the data correlations, such a dense complete graph requires heavy computations and does not scale to a large dataset. Therefore, we choose to further sample from this complete graph to obtain a stochastic binary bipartite graph $\mathbf{G}$ as shown in Figure 3. This graph can be represented as a random binary adjacency matrix, where $\mathbf{G}_{i,j} = 1$ means the reference sequence $\mathbf{x}_i^R$ is a parent of the training sequence $\mathbf{x}_j^M$. We then parameterize this binary adjacency matrix using Bernoulli distributions:

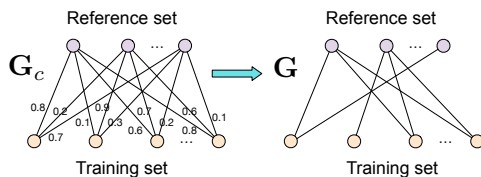

Figure 3: *We sample the (sparse) binary graph $\mathbf{G}$ from the complete weighted (dense) graph $\mathbf{G}_c$.*

$$p(\mathbf{G}|\mathbf{U}_\mathcal{D}) = \prod_{i \in R} \prod_{j \in M} \text{Bernoulli}(\mathbf{G}_{i,j}|\kappa(\mathbf{u}_i^R, \mathbf{u}_j^M)). \tag{5}$$

Intuitively, the edges in $\mathbf{G}_c$ with higher weights are more likely to be kept after sampling. This sampling process leads to sparse correlations for each sampled graph, which can speed up training due to sparsity.

## 3.3 Parameterizing Predictive Distribution

Here we introduce how to parameterize the final prediction based on the three latent variables mentioned in Section 4.1, which *capture the functional uncertainty from different perspectives*.

**Local latent variable $\mathbf{z}_i^M$:** It summarizes the information of the correlated reference points for each training point and captures the *uncertainty of data correlations*. We generate $\mathbf{z}_i^M$ based on the structure of the data correlation graph, and each dimension $k$ follows a Gaussian distribution:

$$\mathbf{z}_{i,k}^M \sim \mathcal{N}(C_i \sum_{j:\mathbf{G}_{j,i}=1} h_1(\mathbf{u}_j^R)_k, \exp(C_i \sum_{j:\mathbf{G}_{j,i}=1} h_2(\mathbf{u}_j^R)_k)), \tag{6}$$

where $h_1$ & $h_2$ are two MLPs and $C_i = \sum_j \mathbf{G}_{i,j}$ is for normalization. As we can see from Equation 6, if the sequence has lower probability to be connected with the reference sequences, $\mathbf{z}_i^M$ becomes a standard Gaussian distribution which is an uninformative prior. This property imposes a similar inductive bias as in the GPs with RBF kernel.

**Global latent variable $\mathbf{v}$.** It encodes the information in *all the reference points*, computed as:

$$\beta_1, \ldots, \beta_{N_R} = \text{Self-Atten}(\mathbf{u}_1^R, \ldots, \mathbf{u}_{N_R}^R), \qquad \mathbf{v} = \sum_{i=1}^{N_R} \beta_i \mathbf{u}_i^R. \tag{7}$$

In contrast with the local variable $\mathbf{z}_i^M$, the global latent variable $\mathbf{v}_i$ summarizes the overall information of the underlying function. It is shared by all the training sequences which allows us to capture the *functional uncertainty from a global level*.

**Sequence embedding $\mathbf{u}_i^M$**: The above two latent variables are both constructed from the embeddings of the reference sequences, which may lose *novel information present in the training sequences*. Therefore, we add a direct path from the latent embedding $\mathbf{u}_i^M$ of the training sequence to the final prediction to enable the neural network to extrapolate beyond the distribution of the reference sequences. This is useful in novel/unprecedented patterns where the input sequence can not rely only on reference sequences from historical data for prediction.

We concatenate the three variables together into a single vector $\mathbf{e}_i$ and obtain the final predictive distribution (where $d_1$ and $d_2$ are MLPs):

$$\mathbf{e}_i = \text{concat}(\mathbf{z}_i, \mathbf{v}_i, \mathbf{u}_i), \qquad p(y_i|\mathbf{z}_i^M, \mathbf{v}, \mathbf{u}_i^M) = \mathcal{N}(d_1(\mathbf{e}_i), \exp(d_2(\mathbf{e}_i))). \qquad (8)$$

### 3.4 Learning the distribution

We now introduce how to learn the model parameters efficiently during training and forecast for a new unseen sequence at test time. Directly maximizing the data likelihood is intractable due to the summation and integral in Equation 1. Therefore, we choose to use the *amortized variational inference* and approximate the true posterior $p(\mathbf{U}_{\mathcal{D}}, \mathbf{G}, \mathbf{Z}_M, \mathbf{v}|R, M)$ with $q_\phi(\mathbf{U}_{\mathcal{D}}, \mathbf{G}, \mathbf{Z}_M, \mathbf{v}|R, M)$, similar to [26], as

$$q_\phi(\mathbf{U}_{\mathcal{D}}, \mathbf{G}, \mathbf{Z}_M, \mathbf{v}|R, M) = p_\theta(\mathbf{U}_{\mathcal{D}}|\mathbf{X}_{\mathcal{D}})p(\mathbf{G}|\mathbf{U}_{\mathcal{D}})p(\mathbf{v}|\mathbf{U}_R)q_\phi(\mathbf{Z}_M|M). \qquad (9)$$

We design $q_\phi$ as a single layer of neural network parameterized by $\phi$, which outputs mean and variance of the Gaussian distribution $q_\phi(\mathbf{Z}_M|\mathbf{X}_M)$.

We then use a gradient-based method, such as Adam [18], to maximize the evidence lower bound (ELBO) of the log likelihood. After canceling redundant terms, the ELBO can be written as:

$$\begin{aligned}
\mathcal{L} = -\mathrm{E}_{\mathbf{Z}_M, \mathbf{G}, \mathbf{U}_{\mathcal{D}}, \mathbf{v} \sim q_\phi(\mathbf{Z}_M|\mathbf{X}_M)p_\theta(\mathbf{G}, \mathbf{U}_{\mathcal{D}}, \mathbf{v}|\mathcal{D})}[\log P(\mathbf{y}_M|\mathbf{Z}_M, \mathbf{U}_M, \mathbf{v}) \\
+ \log P(\mathbf{Z}_M|\mathbf{G}, \mathbf{U}_R) - q_\phi(\mathbf{Z}_M|\mathbf{X}_M)].
\end{aligned} \qquad (10)$$

We use the reparameterization trick to make the sampling procedure from the Gaussian distribution differentiable. Moreover, as sampling from the Bernoulli distribution in Equation 7 leads to discrete correlated data points, we make use of the Gumbel softmax trick [15] to make the model differentiable.

At test time, with the optimal parameter $\theta_{\text{opt}}$, we base the predictive distribution of a new unseen partial sequence $\mathbf{x}^*$ on the reference set as:

$$\begin{aligned}
p(y^*|R, \mathbf{x}^*) = & p_{\theta_{\text{opt}}}(\mathbf{U}_R, \mathbf{u}^*|\mathbf{X}_M, \mathbf{x}^*)p(\mathbf{a}^*|\mathbf{U}_R, \mathbf{u}^*) \\
& p_{\theta_{\text{opt}}}(\mathbf{z}^*|\mathbf{a}^*, \mathbf{U}_R, \mathbf{u}^*)p_{\theta_{\text{opt}}}(y^*|\mathbf{u}^*, \mathbf{z}^*, \mathbf{v})d\mathbf{U}_R d\mathbf{z}^* d\mathbf{v},
\end{aligned} \qquad (11)$$

where $\mathbf{a}^*$ is the binary vector that denotes which reference sequences are the parents of the new sequence. $\mathbf{u}^*$ and $\mathbf{z}^*$ are latent embedding and local latent variable for the new sequence, respectively.

## 4 Experiments

All experiments were done on an Intel i5 4.8 GHz CPU with Nvidia GTX 1650 GPU. The model typically takes around 20 minutes to train. The code is implemented using Pytorch and will be released for research purposes. Supplementary contains additional details and results (e.g. hyperparameters, results on additional metrics (MAPE), additional case and ablation studies).

**Dataset:** In our experiments, we focus on flu forecasting. The CDC uses the ILINet surveillance system to gather flu information from public health labs and clinical institutions across the US. It releases weekly estimates of *weighted influenza-like illness* (wILI)[1]: out-patients with flu-like symptoms aggregated for US national and 10 different regions (called HHS regions). Each flu season begins at week 21 and ends on week 20 of the next year e.g. Season 2003/04 begins on week 21 of 2003 and ends on week 20 of 2004. Following the guidelines of CDC flu challenge [1, 34], we predict from week 40 till the end of season next year. We evaluate our approach using wILI data of 17 seasons from 2003/04 to 2019/20 .

---

[1]`https://www.cdc.gov/flu/weekly/flusight/index.html`

**Goals:** Our experiments were designed evaluate the following. **Q1:** Accuracy and calibration of EPIFNP's forecasts. **Q2:** Importance of different components of EPIFNP. **Q3:** Utility of uncertainty estimates for other related tasks?. **Q4:** Adaptability of EPIFNP to novel behaviors during real-time forecasting. **Q5:** Explainability of predictions.

**Evaluation metrics:** Let $x_{N+1}^{1\ldots t}$ be a given partial wILI test sequence with observed ground truth $y_{N+1}^{(t)}$ i.e., for a $k$-week-ahead task $y_{N+1}^{(t)}$ is just $x_{N+1}^{(t+k)}$. For a model/method $M$ let $\hat{p}_{N+1,M}^{(t)}(Y)$ be the output distribution of the forecast with mean $\hat{y}_{N+1,M}^{(1\ldots t)}$. To measure the predictive accuracy, we use **Root Mean Sq. Error** (RMSE), **Mean Abs. Per. Error** (MAPE) and **Log Score** (LS) which are commonly used in CDC challenges [1, 34]). To evaluate the calibration of the predictive distribution we introduce a new metric called **Calibration Score** (CS). For a model $M$ we define a function $k_M : [0, 1] \to [0, 1]$ as follows. For each value of confidence $c \in [0, 1]$, let $k_M(c)$ denote the fraction of observed ground truth that lies inside the $c$ confidence interval of predicted output distributions of $M$. For a perfectly calibrated model $M^*$ we would expect $k_{M^*}(c) = c$. CS measures the deviation of $k_M$ from $k_{M^*}$. Formally, we define CS as:

$$CS(M) = \int_0^1 |k_M(c) - c| dc \approx 0.01 \sum_{c \in \{0, 0.01, \ldots, 1\}} |k_M(c) - c|. \tag{12}$$

For all metrics, lower is better. We also define the **Calibration Plot** (CP) as the profile of $k_M(c)$ vs $c$ for all $c \in [0, 1]$.

**Baselines:** We compare EPIFNP with standard and state-of-art models used for flu forecasting before, as well as methods typically used for learning calibrated uncertainty quantification.
*Flu forecasting related:* • **SARIMA:** Seasonal Autoregressive Integrated Moving-Average is a auto-regressive time series model used as baseline for forecasting tasks [1, 44]. • **Gated Recurrent Unit** (GRU): A popular deep learning sequence encoder, used before as a baseline for this problem [1]. • **Empirical Bayes** (EB): Utilizes a bayes framework and has won few epidemic forecasting competitions in past [4]. • **Delta Density** (DD): A probabilistic modelling approach that learns distribution of change in successive wILI values given changes from past weeks [5]. • **Epideep** (ED) [1]: Recent state-of-the-art NN flu prediction model based on learning similarity between seasons. • **Gaussian Process** (GP) [44]: Recently proposed statistical flu prediction model using GPs. Note that ED, SARIMA and GRU can only output point estimates and we use the ensemble approach to obtain their uncertainty estimates following [34, 6].
*General ML Uncertainty related:* • **Monte Carlo Dropout** (MCDP) [10]: MCDP applies dropout at testing time for multiple times to measure the uncertainty. We use MCDP on a GRU as a baseline. • **Bayesian neural network** (BNN) [3]: BNN imposes and learns from probability distributions over model parameters. We used LSTM as the architecture for BNN • **Recurrent Neural Process** (RNP) [31]: This method builds on Neural Process framework to learn from sequential data.

*Note:* We need to train EPIFNP only once at start of a season using data from all past seasons unlike some baselines (ED, EB, GP, SARIMA, DD) which require retraining each week.

### 4.1 Q1 & Q2: Forecast Accuracy, Calibration and Model Ablation

Table 1: Average US National Performance: $k$ week ahead forecasting for seasons 2014/15-2019/20.

| | RMSE | | | MAPE | | | LS | | | CS | | |
|---|---|---|---|---|---|---|---|---|---|---|---|---|
| *Model* | k=2 | k=3 | k=4 | k=2 | k=3 | k=4 | k=2 | k=3 | k=4 | k=2 | k=3 | k=4 |
| **ED** | 0.73 | 1.13 | 1.81 | 0.14 | 0.23 | 0.33 | 4.26 | 6.37 | 8.75 | 0.24 | 0.15 | 0.42 |
| **GRU** | 1.72 | 1.87 | 2.12 | 0.28 | 0.31 | 0.356 | 7.98 | 8.21 | 8.95 | 0.16 | 0.2 | 0.22 |
| **MCDP** | 2.24 | 2.41 | 2.61 | 0.46 | 0.51 | 0.6 | 9.62 | 10 | 10 | 0.24 | 0.32 | 0.34 |
| **GP** | 1.28 | 1.36 | 1.45 | 0.21 | 0.22 | 0.26 | 2.02 | 2.12 | 2.27 | 0.24 | 0.25 | 0.28 |
| **BNN** | 1.89 | 2.05 | 2.43 | 0.34 | 0.46 | 0.51 | 6.92 | 7.56 | 8.03 | 0.18 | 0.22 | 0.25 |
| **SARIMA** | 1.43 | 1.81 | 2.12 | 0.28 | 0.35 | 0.42 | 3.11 | 3.4 | 3.81 | 0.43 | 0.38 | 0.34 |
| **RNP** | 0.61 | 0.98 | 1.18 | 0.13 | 0.22 | 0.29 | 3.34 | 3.61 | 3.89 | 0.43 | 0.46 | 0.45 |
| **EB** | 1.21 | 1.23 | 1.25 | 0.57 | 0.58 | 0.58 | 6.92 | 7 | 7.12 | 0.07 | 0.082 | 0.085 |
| **DD** | 0.6 | 0.79 | 0.94 | 0.35 | 0.41 | 0.45 | 3.56 | 3.87 | 4.02 | 0.12 | 0.12 | 0.13 |
| **EPIFNP** | **0.48** | **0.79** | **0.78** | **0.089** | **0.128** | **0.123** | **0.56** | **0.84** | **0.89** | **0.068** | **0.081** | **0.035** |

**Prediction Accuracy:** We first compare the accuracy of EPIFNP against all baselines for real-time forecasting in Table 1. EPIFNP *significantly* outperforms all other baselines for RMSE, MAPE, LS (which measure forecast accuracy). We notice around 13% and 42% improvement over the second

best baseline in RMSE and MAPE respectively. Impressively LS of EPIFNP is *2.5 to 3.5* times less than closest baseline[2]. This is because the intervals $y_i^{(t)} \pm 0.5$ of ground truth consistently fall inside high probability regions of our forecast distribution due to better accuracy (of mean) in general. Even during weeks of uncertainty (like around the peaks) most baselines badly calibrated forecasts don't sufficiently cover the interval, EPIFNP's distribution are wide enough to capture this interval thanks to its superior representation power. We also observed similar results for the 10 HHS regions as well where EPIFNP outperforms the baselines where we show 16% and 7% improvements in RMSE ans LS respectively showing EPIFNP's proficiency over large variety different regions and seasons.

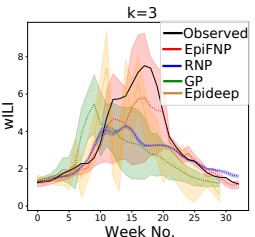

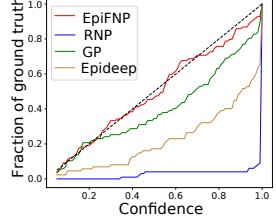

Figure 4: *Forecasts and 95% confidence bounds on 2017/18 season.*

Figure 5: *CPs for EPIFNP and next 3 accurate baselines, k=4*

**Calibration Quality:** We measure how well-calibrated EPIFNP's uncertainty bounds (Figure 4) are via CS. EPIFNP was again the clear winner both for national forecasts (Table 1) and regional forecasts. Calibration Plots (CPs) (Figure 5) show EPIFNP is much closer to the diagonal line (ideal calibration) compared to even the most competitive baselines. We also observed that applying post-hoc calibration methods [23, 39] doesn't effect the significance of EPIFNP's calibration performance (Appendix Table 4). EPIFNP *is clearly significantly superior to all other baselines in predicting both a better calibrated and more accurate forecast distribution.*

**Ablation studies:** We found all three of our EPIFNP components important for performance, with the data correlation graph the most relevant in determining uncertainty bounds. Refer to supplementary for complete results and further discussion.

## 4.2 Q3: Effective uncertainty estimates: Autoregressive inference

**Motivation:** We further show the usefulness and quality of our uncertainty estimates by leveraging the so-called 'auto-regressive' inference (ARI) task. It is common to perform such forecasting in real-time epidemiological settings, especially as accuracy and training data typically drops with increasing $k$ week-ahead in future [37]. In this task, the model uses its own output for $k = 1$ forecast as input (multiple samples) to predict $k = 2$ forecasts and so on to derive $k$-week ahead prediction. Hence an inaccurate and badly calibrated initial model's forecasts propagate their errors to subsequent predictions as well.

We perform forecasting for $k = 2, 3, 4$ week ahead as described above using the $k = 1$ trained model. The pseudocode for Autoregressive inference is given in the Appendix.

Table 2: Evaluation scores for ARI task.

| Model | RMSE | | | LS | | | CS | | |
|---|---|---|---|---|---|---|---|---|---|
| | k=2 | k=3 | k=4 | k=2 | k=3 | k=4 | k=2 | k=3 | k=4 |
| ED | 2.21 | 3.13 | 3.82 | 6.03 | 8.84 | 10 | 0.42 | 0.45 | 0.48 |
| MCDP | 3.62 | 4.03 | 4.39 | 10 | 10 | 10 | 0.47 | 0.46 | 0.49 |
| BNN | 3.41 | 4.23 | 4.78 | 10 | 10 | 10 | 0.39 | 0.41 | 0.42 |
| GP | 1.24 | 1.31 | 1.38 | 4.62 | 5.17 | 5.51 | 0.37 | 0.36 | 0.37 |
| EPIFNP | **0.6** | **0.85** | **0.99** | **0.64** | **0.96** | **1.14** | **0.063** | **0.074** | **0.048** |

**Results:** See Table 2. Only baselines not trained autoregressively by default (as EPIFNP already outperforms them (Q1)) are considered. EPIFNP outperforms all and is comparable even to the *non AR trained original* EPIFNP scores (Table 1) whereas we observed a significant deterioration in scores for other baselines, as anticipated.

## 4.3 Q4: Reacting to abnormal/novel patterns

**Motivation:** A major challenge in real-time epidemiology [36] is the presence of novel patterns e.g. consider the impact of the COVID-19 pandemic on the 2019/20 wILI values (see Figure 1a). In such cases, a trustworthy real-time forecasting model to anticipate, quantify and adapt is needed to such abnormal situations. We studied our performance for the 2009/10 and 2019/20 seasons, which are well known abnormal seasons (due to the H1N1 strain and the COVID-19 pandemic respectively).

---

[2]Our results are statistically significant using the Wilcox signed ranked test ($\alpha = 0.05$) with 5 runs.

While we discuss results for $k = 3$ week ahead forecasting of 2019/20 season, the results for 2009/10 season and for $k = 1, 2, 4$ lead to similar conclusions.

**Results:** In short, EPIFNP *reacts reliably and adapts to novel scenarios.* EPIFNP outperforms other baselines in all metrics. We observed 18% and 31% reduction in RMSE and MAPE respectively compared to best baseline (RNP) and 3.7 times lower LS compared to best baseline (GP). Figure 6(a) shows the prediction and uncertainty bounds of EPIFNP and top 2 baselines. GP and most other baselines (except RNP) fail to capture the unprecedented third peak around week 24. Calibration Plot in Figure 6(b) shows that EPIFNP is better calibrated.

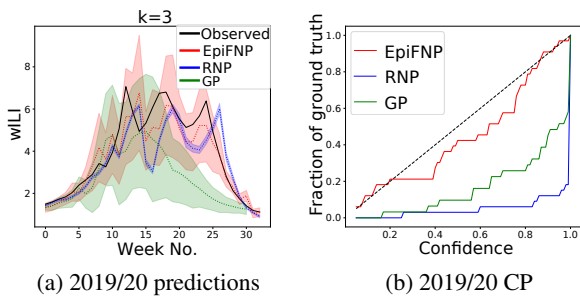

(a) 2019/20 predictions     (b) 2019/20 CP

Figure 6: EPIFNP *outperforms top 2 baselines during abnormal COVID-19 season 2019/20.*

### 4.4  Q5: Explainable Predictions

**Motivation:** Lack of explainability is a major challenge in many ML models, which becomes even more acute in critical domains like public health. Since the Stochastic data correlation graph (SDCG) of EPIFNP (recall Section 3.2) explicitly learns to relate each test sequence with relevant historical seasons' sequences, we can leverage this to provide useful explanations for predictions and model uncertainty. Knowing which past seasons are similar is very helpful for epidemiological understanding of the prevalent strain behavior [1]. We sample SDCGs multiple times and compute average edge probability for every edge between each given historical season and test sequences during real-time forecasting for all weeks. We perform this for $k = 3$ weeks ahead forecasting on season 2015/16 but the observations hold for other seasons and $k = 1, 2, 4$ too.

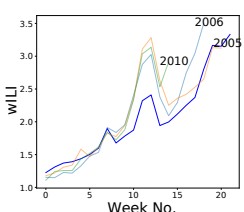 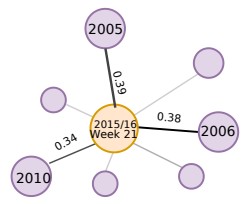

Figure 7: *2015/16 snippet & most similar seasons chosen by* EPIFNP.

Figure 8: *Average edge probabilities for week 21 of 2015/16 season.*

*Obs 1:* EPIFNP *automatically chooses most similar historical seasons relevant at time of prediction.*

We leverage the edge probabilities from the SDCG to examine the seasons that are more likely sampled at at each week. We observed that the seasons with higher probabilities showed similar patterns to that of the current test sequence. Consider week 21 of season 2015/16 during 3 weeks ahead forecasting. The most likely sampled seasons are 2005, 2006 and 2010 (Figure 8). Figure 7 shows these seasons and 2015/16

snippet; clearly they have very similar wILI patterns.

*Obs 2: EpiFNP explains uncertainty bounds of predictions via distribution probabilities in the SDCG.*

As seen in Section 4.3, EPIFNP reacts reliably to abnormal situations and changing trends (e.g. around peaks) by producing larger uncertainty bounds around those events. For example, in Figure 9, uncertainty estimates around peak weeks 12 and 22 are higher than for rest of the weeks. To examine the source of changing uncertainty bounds of prediction, we look at average edge probabilities generated in SDCG (Figure 10) and find that around the peak weeks the edge probabilities are lower than in surrounding weeks. This promotes larger variety of small subsets of the reference set to be sampled during inference that increases the variance of local latent variable $\mathbf{z}_i^M$ thereby increasing the variance of the output forecast distribution.

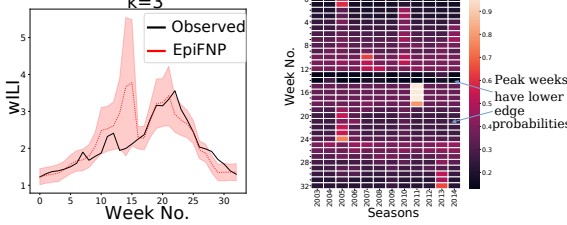

Figure 9: *Higher uncertainty around peaks*

Figure 10: *SDCG Avg. edge probabilities*

# 5  Conclusion

We introduced EPIFNP, a novel deep probabilistic sequence modeling method which generates *well calibrated*, *explainable* and *accurate* predictions. We demonstrated its superior performance in the problem of real-time influenza forecasting by significantly outperforming other non-trivial baselines (more than 2.5x in accuracy and upto 2.4x in calibration). Importantly, it was the only one capable of reliably handling unprecedented scenarios e.g. H1N1 and COVID19 seasons. We also showcased its explainability as it automatically retrieves the most relevant historical sequences matching its current week's predictions using the SDCG. All these highlight the usefulness of EPIFNP for the complex challenge of trustworthy epidemiological forecasting, which directly impacts public health policies and planning. However EPIFNP can be affected by any systematic biases in data collection (for example, some regions might have poorer surveillance and reporting capabilities). There is limited potential for misuse of our algorithms and/or data sources though the dataset is public/anonymized without any sensitive patient information.

We believe our work opens up many interesting future questions. Our setup can be easily extended to handle other diseases and our core technique can be adapted for other general sequence modeling problems. Further, we can extend EPIFNP to also use heterogeneous data from multiple sources. We can also explore incorporating domain knowledge of prior dependencies between different sources/features (e.g. geographically close regions are more likely to have similar disease trends).

**Acknowledgments:** We thank the anonymous reviewers for their useful comments. This work was supported in part by the NSF (Expeditions CCF-1918770, CAREER IIS-2028586, RAPID IIS-2027862, Medium IIS-1955883, Medium IIS-2106961, CCF-2115126, Small III-2008334), CDC MInD program, ORNL, ONR MURI N00014-17-1-2656, faculty research awards from Facebook, Google and Amazon and funds/computing resources from Georgia Tech.

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
