# Appendix for When in Doubt: Neural Non-Parametric Uncertainty Quantification for Epidemic Forecasting

Code for EPIFNP and wILI dataset is publicly available [3].

## A  Additional Related work

**Statistical models for Epidemic Forecasting** In the recent years, statistical models have been the most successful in several forecasting targets, as noted in multiyear assessments [34]. In influenza forecasting, various recent statistical approaches have been proposed. On one hand, we have models designed to model the details on the underlying generative distribution of the data. Among these, [4] proposed a semiparametric Empirical Bayes framework that constructs a prior of the current season's epidemic curve from the past seasons and outputs a distribution over epidemic curves. [5] opts for a non-parametric approach based on kernel density estimation to model the probability distribution of the change between consecutive predictions. Closely related, Gaussian processes have been recently explored for influenza forecasting [44]. Other popular methods rely on ensembles of mechanistic and statistical methods [32].

More recently, the deep learning community has take interest in forecasting influenza [1, 42] and COVID-19 [37]. Indeed, deep learning enables to address novel situations where traditional influenza models fail such as adapting a historical influenza model to pandemic [36]. Deep learning is also suitable because it provides the capability of ingesting data from multiple sources, which better informs the model of what is happening on the ground. However, for most of this body of work uncertainty quantification is either non existent or has been explored with simple techniques that lack of proper knowledge representation. Our work aims to close this gap in the literature.

**Uncertainty Quantification for Deep Learning** Recent works have shown that deep neural networks are over-confident in their predictions [12, 20]. Existing approaches for uncertainty quantification can be categorized into three lines. The first line is based on Bayesian Neural Networks (BNNs) [28, 3, 27]. They are realized by first imposing prior distributions over neural network parameters, then infer parameter posteriors and further integrate over them to make predictions. However, as exact inference of parameter posteriors is often intractable, approximation methods have also been proposed, including variational inference [3, 27], Monte Carlo dropout [10] and stochastic gradient Markov chain Monte Carlo (SG-MCMC) [25, 43]. Such BNN approximations tend to underestimate the uncertainty [21]. Moreover, specifying parameter priors for BNNs is challenging because the parameters of DNNs are huge in size and uninterpretable [21, 26].

The second line tries to combine the stochastic processes and DNNs. Neural Process (NP) [11] defines a distribution over a global latent variable to capture the functional uncertainty, while Functional neural process (FNP) [26] use a dependency graph to encode the data correlation uncertainty. However, they are both for the static data. Recently, recurrent neural process (RNP) [31, 17] has been proposed to incorporate RNNs into the NP to capture the ordering sequential information.

The third line is based on model ensembling [24] which trains multiple DNNs with different initializations and use their predictions for uncertainty quantification. However, training multiple DNNs require extensive computing resources.

## B  Model Hyperparameters

We describe all the hyperparameters used for the EPIFNP model including the model architecture. In general, we used the hyperparameters as done in [26] with changes made to accommodate the sequential modules and global embedding for our use case.

### B.1  Architecture

#### B.1.1  Probabilistic Neural Sequence Encoder

The GRU for the encoder model has single hidden layer of 50 units and outputs 50 dimensional vectors. The Attention layer was similar to that used in transformers. We used a single attention head

---

[3]Link to code and dataset: `https://github.com/AdityaLab/EpiFNP`

and retained the same number of dimensions, 50, when generating the key and value embeddings. to generate $\mathbf{u}_i$, we derived the mean and log variance using a stack of 3 linear layers for $g_1$ and $g_2$ with ReLU in between the hidden layers. All hidden layers have 50 units.

Note that for sampling from multivariate gaussian distribution, we always assumed the covariance matrix to be a diagonal matrix and only derived log variance for each dimension.

### B.1.2  Parameterizing Predictive Distribution

The $h_1$ and $h_2$ functions used to derive $\mathbf{z}_i^M$ were single linear layers with no activation function. The Attention layer used to derive $\mathbf{v}$ was similar to that used in encoder: 1 attention head with 50 dimension units for key and value transforms. $d_1$ and $d_2$ are two modules of feed forward layers with a ReLU function between them with first layer having 50 units and the second having 2 to output mean and log variance of forecast output.

### B.2  Other Hyperparameters

Learning rate used was $1e-4$. We also used early stopping to prevent overfitting and randomly sampled 5% of training points as validation set to determine when we reached the point of overfitting. EPIFNP usually 2000-3000 epoch to complete training. We found that our model was very robust to small changes in architecture and learning rate and we mostly optimized for faster rate of convergence during training.

## C  Details on Evaluation metrics

Let $x_{N+1}^{1\ldots t}$ be a given partial wILI test sequence with observed ground truth $y_{N+1}^{(t)}$ i.e., for a $k$-week-ahead task $y_{N+1}^{(t)}$ is just $x_{N+1}^{(t+k)}$. For a model/method $M$ let $\hat{p}_{N+1,M}^{(t)}(Y)$ be the output distribution of the forecast with mean $\hat{y}_{N+1,M}^{(1\ldots t)}$. Then we define the evaluation metrics as follows. We evaluate all the methods based on metrics for measuring prediction accuracy (RMSE, MAPE and LS are commonly used in CDC challenges [1, 34]) as well as targeted ones (CS) measuring the quality of prediction *calibration* of uncertainty. For all metrics, lower is better. EPIFNP is carefully designed to generate both accurate and well calibrated forecasts, unlike past work which focuses typically on accuracy only.

- **Root Mean Sq**. Error $\mathbf{RMSE}(M) = \sqrt{\frac{1}{T}\sum_{t=1}^{T}(y_{N+1}^{(t)} - \hat{y}_{N+1,M}^{(t)})^2}$

- **Mean Abs. Per. Error** $\mathbf{MAPE}(M) = \frac{1}{T}\sum_{t=1}^{T}\frac{|y_{N+1}^{(t)} - \hat{y}_{N+1,M}^{(t)}|}{|y_{N+1}^{(t)}|}$

- **Log Score** (LS): This score used by the CDC caters to the stochastic aspect of forecast prediction [34].

$$LS(M) = \sum_{i=1}^{T}\frac{1}{T}\int_{y_i^{(t)}-0.5}^{y_i^{(t)}+0.5} -\log(\hat{p}_{N+1,M}^{(t)}(y))dy \qquad (13)$$

The integral is approximated by samples from $\hat{p}_{N+1,M}^{(t)}(Y)$ and calculating the fraction of samples that fall in the correct interval.

- **Calibration Score** (CS): In order to evaluate the calibration of output distribution we introduce a new metric called Calibration Score, which is inspired by reliability diagrams [29] used for binary events. The idea behind the calibration score is that *a well calibrated model provides meaningful confidence intervals*. For a model $M$ we define a function $k_M : [0,1] \to [0,1]$ as follows. For each value of confidence $c \in [0,1]$, let $k_M(c)$ denote the fraction of observed ground truth that lies inside the $c$ confidence interval of predicted output distributions of $M$. For a perfectly calibrated model $M^*$ we would expect $k_{M^*}(c) = c$. CS measures the deviation of $k_M$ from $k_{M^*}$. Formally, we define CS

as:

$$CS(M) = \int_0^1 |k_M(c) - c| dc \approx 0.01 \sum_{c \in \{0, 0.01, ..., 1\}} |k_M(c) - c| \qquad (14)$$

(since integrating over all values of $c$ is intractable in general).

We also define the **Calibration Plot** (CP) as the profile of $k_M(c)$ vs $c$ for all $c \in [0, 1]$.

## D  Detailed forecast results

### D.1  Regional forecasts

We also evaluate our model and baselines on wILI dataset specific to different regions in USA. The wILI data for 10 HHS regions are available separately and each of them have different characteristics in their wILI trends which are affected by local climate, population density and other factors. Therefore we train our models on each of the HHS regions seperately and average the scores to produce the results in table 3. EPIFNP outperforms baselines in most baselines. We observed that for 8 of the 10 regions EPIFNP outperforms all models in all the evaluation metrics across 2 to 4 week ahead forecast tasks. Even for the remaining 2 regions EPIFNP shows superior scores in majority of the metrics.

Table 3: Average Evaluation scores of EPIFNP and baselines across all HHS regions. The scores are averaged over seasons 2014-15 to 2019-20 for all 10 HHS regions.

| | RMSE | | | MAPE | | | LS | | | CS | | |
|---|---|---|---|---|---|---|---|---|---|---|---|---|
| Model | 2 | 3 | 4 | 2 | 3 | 4 | 2 | 3 | 4 | 2 | 3 | 4 |
| ED | 0.86 | 1.2 | 1.81 | 0.23 | 0.25 | 0.36 | 2.89 | 2.69 | 3.32 | 0.17 | 0.32 | 0.33 |
| GRU | 1.95 | 2.05 | 2.76 | 0.39 | 0.41 | 0.43 | 4.41 | 4.52 | 4.86 | 0.37 | 0.38 | 0.41 |
| MCDP | 3.01 | 3.36 | 3.41 | 0.58 | 0.548 | 0.68 | 10 | 10 | 10 | 0.38 | 0.39 | 0.47 |
| GP | 0.64 | 0.83 | 0.95 | 0.19 | 0.22 | **0.25** | **0.92** | 1.44 | 1.63 | 0.13 | 0.16 | 0.15 |
| BNN | 2.25 | 2.87 | 3.02 | 0.26 | 0.29 | 0.35 | 8.31 | 9.89 | 10 | 0.38 | 0.42 | 0.46 |
| SARIMA | 1.81 | 2.33 | 2.8 | 0.36 | 0.47 | 0.58 | 3.3 | 3.87 | 4.37 | 0.39 | 0.37 | 0.37 |
| RNP | 0.87 | 0.88 | 1.17 | 0.19 | 0.23 | 0.29 | 9.27 | 9.58 | 9.78 | 0.46 | 0.46 | 0.47 |
| EB | 1.51 | 1.53 | 1.56 | 0.67 | 0.67 | 0.68 | 7.15 | 7.23 | 7.29 | 0.13 | 0.13 | 0.13 |
| DD | 0.84 | 1.05 | 1.22 | 0.44 | 0.49 | 0.55 | 3.51 | 3.77 | 3.91 | **0.11** | **0.11** | **0.12** |
| EPIFNP | **0.55** | **0.7** | **0.89** | **0.17** | **0.19** | 0.26 | 1.41 | **1.54** | **1.81** | 0.15 | **0.11** | 0.13 |

### D.2  Post-hoc calibration methods

We also evaluated effect of post-hoc methods [23, 39] on calibration of prediction distributions of top baselines and EPIFNP. The results are summarized in Table 4. We observe that EPIFNP doesn't benefit much from post-hoc calibration methods due to its already well-calibrated forecasts. However, they improve the calibration scores of other baselines (sometimes at the cost of prediction accuracy). However, EPIFNP is still clearly the best performing model.

## E  Autoregressive inference

We formally describe how to perform autoregressive inference as discussed in Section 4.2 in Algorithm 1.

### E.1  Results

We provided RMSE, LS and CS of AR task in main paper Table 2. See Table 5 for results for AR task that includes MAPE scores. As described in Section 4.2, EPIFNP outperforms baselines in AR tasks and its performance in comparable to EPIFNP scores trained separately for different values of $k$ (Figure 11).

Table 4: Effect of post-hoc calibration on point estimate and calibration scores. Iso and DC are post-hoc methods introduced in [23] and [39] respectively.

| Model | Post-Hoc | RMSE | | | MAPE | | | LS | | | CS | | |
|---|---|---|---|---|---|---|---|---|---|---|---|---|---|
| | | k=2 | k=3 | k=4 | k=2 | k=3 | k=4 | k=2 | k=3 | k=4 | k=2 | k=3 | k=4 |
| **EPIFNP** | None | **0.48** | **0.79** | **0.78** | **0.089** | **0.128** | **0.123** | **0.56** | **0.84** | **0.89** | **0.068** | **0.081** | **0.035** |
| | Iso | **0.49** | **0.81** | **0.79** | **0.09** | **0.124** | **0.119** | **0.56** | **0.86** | **0.9** | **0.08** | **0.09** | **0.07** |
| | DC | **0.44** | **0.74** | **0.77** | **0.088** | **0.114** | **0.117** | **0.55** | **0.75** | **0.86** | **0.07** | **0.08** | **0.035** |
| **RNP** | None | 0.61 | 0.98 | 1.18 | 0.13 | 0.22 | 0.29 | 3.34 | 3.61 | 3.89 | 0.43 | 0.38 | 0.34 |
| | Iso | 1.77 | 2.26 | 2.18 | 0.18 | 0.27 | 0.28 | 2.55 | 2.62 | 3.12 | 0.18 | 0.23 | 0.24 |
| | DC | 1.73 | 2.17 | 2.25 | 0.18 | 0.27 | 0.31 | 1.53 | 1.84 | 2.05 | 0.13 | 0.12 | 0.15 |
| **GP** | None | 1.28 | 1.36 | 1.45 | 0.21 | 0.22 | 0.26 | 2.02 | 2.12 | 2.27 | 0.24 | 0.25 | 0.28 |
| | Iso | 2.24 | 2.51 | 2.72 | 0.34 | 0.34 | 0.38 | 1.97 | 2.13 | 2.16 | 0.094 | 0.12 | 0.11 |
| | DC | 2.15 | 2.68 | 2.72 | 0.32 | 0.37 | 0.39 | 1.94 | 2.07 | 2.04 | 0.09 | 0.11 | 0.1 |
| **EpiDeep** | None | 0.73 | 1.13 | 1.81 | 0.14 | 0.23 | 0.33 | 4.26 | 6.37 | 8.75 | 0.24 | 0.15 | 0.42 |
| | Iso | 1.02 | 1.25 | 1.94 | 0.16 | 0.24 | 0.34 | 2.46 | 4.58 | 4.64 | 0.21 | 0.11 | 0.19 |
| | DC | 1.15 | 1.28 | 1.74 | 0.17 | 0.26 | 0.32 | 2.11 | 3.97 | 3.65 | 0.18 | 0.14 | 0.21 |
| **MCDP** | None | 2.24 | 2.41 | 2.61 | 0.46 | 0.51 | 0.6 | 9.62 | 10 | 10 | 0.24 | 0.32 | 0.34 |
| | Iso | 2.36 | 2.58 | 2.53 | 0.45 | 0.47 | 0.59 | 6.72 | 9.64 | 10 | 0.14 | 0.26 | 0.31 |
| | DC | 2.31 | 2.44 | 2.52 | 0.44 | 0.48 | 0.57 | 6.31 | 8.24 | 10 | 0.15 | 0.22 | 0.25 |

---

**Algorithm 1:** Autoregressive inference (ARI)

---

**Input** : Model $M$ trained for 1 week ahead forecasting, test sequence $x_i^{(1...t)}$, $k$: No. of weeks ahead to forecast

**Output** : Distribution $\hat{P}_M(X_i^{(t+k)}|x_i^{(1...t)})$ for forecasting $x_i^{(t+k)}$

```
/* Z_i is the set of candidate sequences for t + i + 1 forecasting.  Each sequence
   has first t values as x_t^{1...t} and next i values are sampled by ARI        */
```

1   $Z_0 = \{x_i^{(1...t)}\}$;
2   **for** *i in 1 to k* **do**
3      **for** *j in 1 to N* **do**
4         Sample sequence $\bar{x}$ from $Z_{i-1}$;
5         Feed $\bar{x}$ to $M$ and sample output $y$;
6         Append $y$ to $\bar{x}$ to form a new sequence $\bar{x} \oplus \{y\}$;
7         Add $\bar{x} \oplus \{y\}$ to $Z_i$;
```
// x̄ ⊕ {y} is a candidate sequence for t + i + 1 forecast.
```
8      **end**
9   **end**
10   **preds** $= \{x : x$ is last element of $\bar{x} \in Z_k\}$;
11   Approximate $\hat{P}_M(X_i^{(t+k)}|x_i^{(1...t)})$ from **preds**

---

## F   Ablation study

We examine the effectiveness of three components of EPIFNP in learning accurate predictions and good calibration of uncertainty: (1) Global Latent Variable $\mathbf{v}$ , (2) Local latent variable $\mathbf{z}_i^M$ (3) Modelling sequence encodings $\mathbf{u_i}$ as a random variable instead of directly using deterministic encodings $\bar{\mathbf{h}}_i$. Detailed results of this study are in Table 6. All three components are essential for best performance of the model. Removing $\mathbf{z}_i^M$ shows very large decrease in log scores and calibration scores. This aligns with the hypothesis about role of data correlation graph in determining uncertainty bounds (see Section 4.4).

We present the results of ablation experiments in Table 6. We see that all three components are essential for best performance of the model. Removing $\mathbf{z}_i^M$ shows large decrease in log scores and calibration scores as the model becomes less capable of modelling uncertainty. Of all the ablation models, making latent embeddings deterministic seems to have least effect on performance though the reduction is still very detrimental to overall performance.

## G   EPIFNP adapts to H1N1 Flu season

EPIFNP outperforms all baselines and has 30% and 10% better RMSE and MAPE scores compared to second best baseline (RNP). LS of EPIFNP is 0.48, about *9.8 times* lesser than second best model. Figure 12(a) shows the prediction and 95% confidence bounds of EPIFNP and two best performing

Table 5: Evaluation scores for ARI task (Section 4.2)

| Model | RMSE | | | MAPE | | | LS | | | CS | | |
|---|---|---|---|---|---|---|---|---|---|---|---|---|
| | $k=2$ | $k=3$ | $k=4$ | $k=2$ | $k=3$ | $k=4$ | $k=2$ | $k=3$ | $k=4$ | $k=2$ | $k=3$ | $k=4$ |
| ED | 2.21 | 3.13 | 3.82 | 0.4 | 0.43 | 0.55 | 6.03 | 8.84 | 10 | 0.42 | 0.45 | 0.48 |
| MCDP | 3.62 | 4.03 | 4.39 | 0.58 | 0.61 | 0.67 | 10 | 10 | 10 | 0.47 | 0.46 | 0.49 |
| BNN | 3.41 | 4.23 | 4.78 | 0.51 | 0.55 | 0.62 | 10 | 10 | 10 | 0.39 | 0.41 | 0.42 |
| GP | 1.24 | 1.31 | 1.38 | 0.21 | 0.21 | 0.24 | 4.62 | 5.17 | 5.51 | 0.37 | 0.36 | 0.37 |
| EPIFNP | **0.6** | **0.85** | **0.99** | **0.1** | **0.14** | **0.166** | **0.64** | **0.96** | **1.14** | **0.063** | **0.074** | **0.048** |

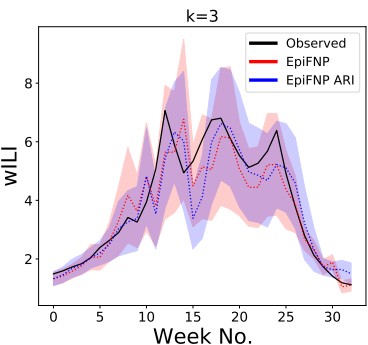

Figure 11: Uncertainty bounds of ARI EPIFNP and normally trained EPIFNP are similar.

baselines. EPIFNP captures the unprecedented early peak observed around week 4. There is also a high uncertainty bounds around the peak. In contrast RNP has very small uncertainty bounds. GP and most other baselines (except GRU, RNP and MCDP) do not even capture the peak. Calibration plot in Figure 12(b) shows the deviation of EPIFNP from ideal diagonal to be much smaller compared to other baselines. This results in about *4.6 times* smaller CS compared to the best baseline.

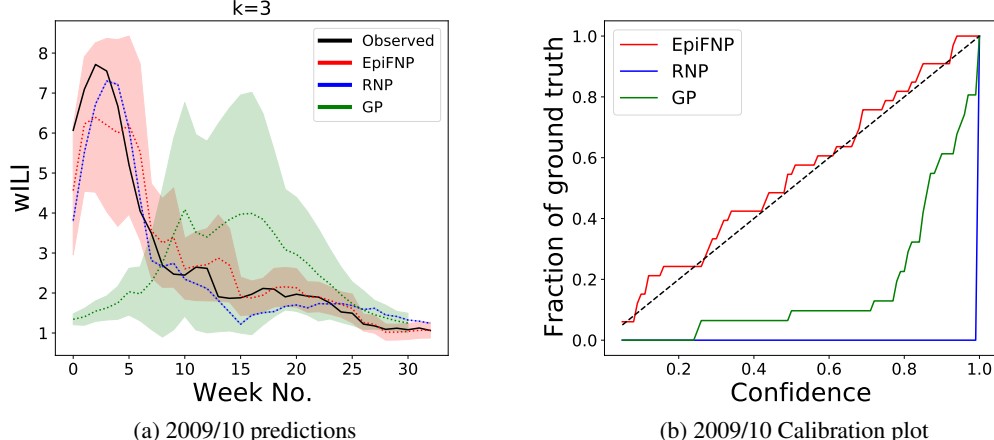

(a) 2009/10 predictions

(b) 2009/10 Calibration plot

Figure 12: EPIFNP outperforms baselines on real-time forecasting during abnormal H1N1 season (2009/10). Forecasts for $k = 3$ weeks ahead forecast by EPIFNP and next two best baselines: RNP and GP.

Table 6: Ablation study to measure the effects of 1) Local latent variable $\mathbf{z}_i^M$ 2) Global latent variable $\mathbf{v}$ and 3) Stochastic SeqEncoder: Modelling $\mathbf{u_i}$ as stochastic latent variables rather than deterministic encodings.

| Ablation study | RMSE | | | MAPE | | |
|---|---|---|---|---|---|---|
| Model/Weeks ahead | 2 | 3 | 4 | 2 | 3 | 4 |
| EPIFNP | **0.48** | **0.79** | **0.78** | **0.089** | **0.128** | **0.123** |
| -(Local latent variable) | 0.99 | 1.45 | 1.51 | 0.17 | 0.25 | 0.29 |
| -(Global latent variable) | 1.76 | 2.05 | 2.45 | 0.33 | 0.41 | 0.42 |
| - (Stochastic Encoder) | 0.87 | 1.09 | 1.19 | 0.15 | 0.21 | 0.22 |
| -(Stochastic Encoder, Local latent variable) | 1.18 | 1.39 | 1.83 | 0.17 | 0.18 | 0.21 |
| - (Stochastic Encoder, Global latent variable) | 0.67 | 0.73 | 0.9 | 0.19 | 0.2 | 0.26 |
| Ablation study | LS | | | CS | | |
| Model/Weeks ahead | 2 | 3 | 4 | 2 | 3 | 4 |
| EPIFNP | **0.51** | **0.78** | **1.2** | **0.069** | **0.081** | **0.035** |
| -(Local latent variable) | 3.51 | 6.67 | 8.09 | 0.21 | 0.27 | 0.29 |
| -(Global latent variable) | 2.06 | 2.41 | 3.37 | 0.085 | 0.12 | 0.19 |
| - (Stochastic Encoder) | 3.13 | 3.53 | 4.88 | 0.14 | 0.19 | 0.24 |
| -(Stochastic Encoder, Local latent variable) | 6.11 | 8.91 | 9.68 | 0.44 | 0.48 | 0.47 |
| - (Stochastic Encoder, Global latent variable) | 2.21 | 3.58 | 3.72 | 0.41 | 0.45 | 0.42 |