# OpenReview forum: "When in Doubt: Neural Non-Parametric Uncertainty Quantification for Epidemic Forecasting"
_NeurIPS.cc/2021/Conference — NeurIPS 2021 Poster_

### Official Review · Reviewer_wNUL · 2021-07-11

**Rating:** 7
**Confidence:** 4

**Summary:**

The paper develops a deep probabilistic model for forecastic stochastic time series with application to epidemic forecasting. The proposed algorithm encodes data points (weekly flu records) comprising a time series of arbitary size into a fixed-sized embedding by passing it through a GRU and binding the hidden states via self attention. Another round of self attention is used to relate whole time series (seasons). Next, it calculates a probabilistic cross attention graph, the edges of which match the whole time series of full seasons of past years to snippets from the current season. The data generating process where all building blocks come together is inferred by a standard variant of amortized variational Bayes. The paper conducts a rigorous set of experiments on a public flu statistics database to analyze multiple aspects of the proposed method and the benchmarks, such as prediction accuracy, uncertainty calibration, and explainability. The proposed method improves the state of the art with respect to all these aspects with a clear margin.

**Ethical Concerns:**

The paper touches the ethical implications of the introduced method only in Line 408 at a rather superficial level. I do not agree with the statement that the potential misuse of their algorithm is too limited to raise awareness about it. The same algorithms can well also be used for prediction of other types of time series, such as crime or credit return forecasting, where the fairness of the model's prediction could be a matter of concern for its users.

**Limitations And Societal Impact:**

The paper does not provide an investigation of the limitations of the proposed algorithm. The only weakness mentioned in Section 5 is a feature of data, not the algorithm.

**Main Review:**


Significance of Novelty:

The studied application, epidemic forecasting, is obviously impactful. From the applied research standpoint, the methodological contribution of the proposed algorithm is sufficiently novel, as it is carefully engineered towards epidemics data. From the fundamental machine learning research standpoint, it is rather incremental. One can view the proposed method as an "Attentive Neural Process" [Rv1], with the cross attention part replaced by the random graph used in the "Functional Neural Process" design [22]. The inference algorithm is also a direct adaptation from [22].

Technical Flaws:

The paper is technically fully solid. Despite the proposed solution consists of a rather complex network of multiple compartments, they follow intelligible design choices.

Presentation:

The presentation clarity of the paper is sufficient, though not outstanding. The content has a few circular dependencies.

i) The variable "v" used in Eq 1 does not mean much, given the introduction in Line 143. It becomes much clearer in Eq 7, but at the cost of going back to Eq 1 once more. Its introduction in 143 could be detailed a little more to justify its necessity.

ii) Figure 2 is too complicated to be descriptive. I recommend extending the caption with more details about the essential properties and novel aspects of the building blocks of the proposed method.

iii) Figure 4 to 9 are way too small to be readable without zooming in electronically. Some further material (or a subset of these figures) could be moved to the supplement and the remaining can be presented at least double as big as their current size.

iv) [MINOR] Even though the paper cites the "Recurrent Attentive Neural Process" [26], perhaps it is better to cite also the seed "Attentive Neural Process" paper [Rv1]

Experimental Evaluation:

The paper conducts a detailed and systematically designed series of experiments that investigates various important aspects of the proposed method, spanning prediction accuracy, uncertainty quantification, and explainability. The method stands out among all the baselines in all experiments by a clear margin.

Overall:

This is solid work that introduces a novel time series forecasting method engineered tediously for epidemic data. The experimental evaluation is also rigorous and the results are very strong. My initial recommendation is an accept.

[Rv1] Kim et al., Attentive Neural Processes, ICLR, 2019

**Time Spent Reviewing:**

3

---

> ### Author Response · Authors · 2021-08-10
> **Response to Reviewer 3**
>
> We thank the reviewer for their valuable comments.
>
> **Figure 2 is too complicated to be descriptive. I recommend extending the caption with more details about the essential properties and novel aspects of the building blocks of the proposed method.** **Figure 4 to 9 are way too small to be readable without zooming in electronically. Some further material (or a subset of these figures) could be moved to the supplement and the remaining can be presented at least double as big as their current size.**
>
> Thanks for the suggestions. We will make these changes in the final version.
>
> **Even though the paper cites the "Recurrent Attentive Neural Process" [26], perhaps it is better to cite also the seed "Attentive Neural Process" paper**
>
> Thank you for pointing this out. We will add the citation to the ANP paper as well.
>
> **Regarding ethical impact**
>
> We agree that while our method can be applied to general sequence prediction and time-series forecasting problems, this presents opportunities for potential misuse based on the task at hand (Line 408). However, we also believe that improved calibration in forecasts also opens up avenues for fairer predictions [1] and is an interesting line of work to explore. We also mentioned the impact of data quality and availability on forecast quality (Line 206) which can be detected by examining the calibration of predictions. We will mention this line of thought in our final version.
>
>
> ## References
>
> [1] Pleiss, Geoff, et al. "On fairness and calibration." NIPS 2017

---

> > ### Comment · Reviewer_wNUL · 2021-08-16
> > **I keep my score**
> >
> > Thanks for your reply. I did not have major issues to be fixed anyway. Great to know that we also agree on the points of improvement.
> >
> > I have also read the other reviews. Although I agree with many of the mentioned issues there, I do not view them fundamental enough to shadow the originality of the model and the strength of its empirical demonstration on a challenging and impactful real-world application.

---

### Official Review · Reviewer_mtym · 2021-07-15

**Rating:** 6
**Confidence:** 3

**Summary:**

This paper proposes a method for probabilistic time series forecasting for univariate real-valued data (they just consider Influenza-Like Illness) using a semi-parametric neural network. The basic idea is to embed the prefix of observations for the current season into a fixed-sized latent vector using a GRU, and to embed all past full-year sequences into latent vectors, and then to compute a weighted average of the most similar embeddings from the past and to use this feature inside an MLP to predict the future, k-weeks ahead. They evaluate on the CDC dataset and report improved point estimates and calibration scores.

**Limitations And Societal Impact:**

The main limitation is that the method seems over-engineered to one specific dataset (ILI from CDC), and it is not clear how robust or general purpose it is. The focus on uncertainty estimation, however, is good.

**Main Review:**

This paper feels like it's using a sledge hammer to open a walnut: the dataset they are working with is tiny (52 scalars per year, 17 years  / seasons of data), but the model they fit is a very complex semi-parametric neural net, combining GRUs, VAEs, self-attention, and functional neural processes (oh my!). The potential for overfitting is very high!

It is not clear exactly how the authors did their evaluation. It seems (from table 1) they evaluated predictions on years 2014-2019, and therefore presumably trained on 2003-2013, which is fine. But if the reference set is all the sequences in 2003-2013, then wouldn't a simpler baseline like K-nearest neighbors work as well? (e.g.,  I can match the prefix for the current year with the prefix for all past years, using something like dynamic time warping as my similarity metric). It seems like the model is learning to do this kind of similarity matching, but it would be more enlightening to show the behavior of the model on some synthetic data, where we can see where it works and where it fails.

The evaluation metrics seem a bit limited. RMSE, MAPE and Log score are standard.  But I am concerned the authors needed to invent a "new metric" for calibration. (The one they proposed seems standard to me.) There is a large body of work on assessing the predictive accuracy of probabilistic forecasts (see eg Bracher'21 for a recent method),  it would be good to include some of these metrics.

The use of a Gaussian observation model for a time series of positive counts seems inappropriate. A negative-binomial may be a better choice (if integer valued). Or some kind of quantile predictor.

J. Bracher, E. L. Ray, T. Gneiting, and N. G. Reich, “Evaluating epidemic forecasts in an interval format,” PLoS Comput. Biol., 2021 [Online]. Available: http://arxiv.org/abs/2005.12881


**Time Spent Reviewing:**

1

---

> ### Author Response · Authors · 2021-08-10
> **Response to Reviewer 2**
>
> We thank the reviewer for their comments.
>
> **This paper feels .... the dataset they are working with is tiny (52 scalars per year, 17 years / seasons of data), but the model they fit is a very complex semi-parametric neural net, combining ... The potential for overfitting is very high!**
>
>
> We agree our model is an overparameterized model, but this is needed because
> disease forecasting is a challenging problem that requires capturing complex
> patterns (e.g., composite signals, multi-periodic with variable evolving similarities, noise, rise-and-fall, regional level differences, and variations in  viral strains, healthcare seeking behaviors, and reporting patterns) for accurate forecasting. Therefore, simpler time-series models like GRU and SARIMA are not sufficient as shown in our results. Many
> prior attempts to this problem employ complex statistical and mechanistic models
> [1,2,6] with a larger number of parameters than our model (indeed calibrating large agent-based models is still an art and an open research question). Such prior works
> reflect the necessity of using expressive models to handle the difficulty in
> disease forecasting. Most prior works also consider calibration as a
> second-class citizen to accuracy whereas reliable forecasts in practice require
> well-calibrated confidence scores for sound decision making. Our approach aims
> to fulfill this requirement by proposing a unifying framework leveraging
> Gaussian Process-based probabilistic modelling with representation power of deep
> sequential models to provide accurate, well-calibrated and explainable
> forecasts.
>
>
> We do not agree that our model is overfitting the training data. First, our
> experiment is a real-time forecasting setup, which used past years to forecast
> the future. There is no overlap between training and test data, and the test
> data can even contain shifted trends and distributions. Our experimental
> results show that our model has superior performance for such a forecasting
> setup, and even adapts to unseen trends in H1N1 and Covid-19. Such performance
> would not be possible if our model is overfitting the training data, but only
> possible when the model has learned generalizable epidemiological knowledge.
> Second, the well-calibrated prediction is another evidence that our model is not
> overfitting the training dataset. Because overfitting usually causes the model
> to have over-confident predictions on test data.
>
>
> Why over-parameterized deep learning models do not suffer overfitting?  This is still an important open research question in
> deep learning and has been attracting much attention [8,9,10,11,12]. Some progresses towards this question have shown that the
> sample complexity measures in classic generalization theory such as VC-dimension
> and Rademacher complexity are too pessimistic and have proposed tighter
> sample efficiency bounds for deep neural networks; and some works have shown
> that overparameterization in deep learning can help SGD optimization to escape
> poor local optima and converge faster. In addition to such existing results, we
> believe the stochastic correlation graph in our model may have also contributed
> to avoiding overfitting. As our model randomly samples a small subset of
> reference points based on the learned pattern similarity, this has an analogy to
> using dropout for avoiding overfitting as in standard neural networks.
>
> **It is not clear exactly how the authors did their evaluation. It seems (from table 1) they evaluated predictions on years 2014-2019, and therefore presumably trained on 2003-2013, which is fine.**
>
> We like to clarify that, as described in Section 2 we emulate the real-time forecasting setup for our task. We train for each season using data from all previous seasons. For example, to train for the 2018/19 season we use historical data from seasons 2003/04 to 2017/18 to train our model and predict for the current season. Thus, the reference points contain sequences till the 2017/18 season.
>
> **But if the reference set is all the sequences in 2003-2013, then wouldn't a simpler baseline like K-nearest neighbors work as well?**
>
> First, we like to point out that the epidemic forecasting task has non-trivial patterns which dynamically evolve over time. Also, each season can borrow patterns from multiple previous seasons [3] as well as exhibit unseen behaviors.
> Thus, simple baselines like KNN and Histogram based on static time series similarity measures perform poorly, as shown in prior work [2,3].
>
> Second, our method exploits similarity in latent space which can capture more complex patterns. In Sec 4.4, we provide specific examples to show the similar patterns learned by EpiFNP have strong explainability for prediction and uncertainty estimation.
>
> **The evaluation metrics seem a bit limited. RMSE, MAPE and Log score are standard. But I am concerned the authors needed to invent a "new metric" for calibration. There is a large body of work on assessing the predictive accuracy of probabilistic forecasts (see eg Bracher'21 for a recent method), it would be good to include some of these metrics.**
>
> RMSE and MAPE are standard point-estimate metrics used for measuring accuracy in flu forecasting. Log score has been a standard probabilistic metric [2] that is also used in the CDC organized annual Flusight challenge for multiple years. Coming up with metrics for evaluating probabilistic forecasts is still an open research area in this field. The WIS score referred by the reviewer, for example, was only very recently introduced by Bracher'21 and is directly derived from the log score. It has been used only for COVID forecasting yet. Hence, we chose to use the log score instead since it is still a standard metric for flu-forecasting as used by other top models including our baselines.
>
> In contrast to accuracy, due to limitations in the availability of good calibration metrics, we had to introduce the calibration score to measure the calibration and uncertainty of forecasts.
> The calibration score is a direct extension of expected calibration error (ECE) used in classification to a regression setting. It is a simple measure of how close confidence scores reported by our forecasts relate to the actual fraction of ground truth points in a given interval. A similar score was also proposed in [5] and we will cite this paper in the final version.
>
> **The use of a Gaussian observation model for a time series of positive counts seems inappropriate. A negative-binomial may be a better choice (if integer valued). Or some kind of quantile predictor.**
>
> Negative-binomial distribution does not work since wILI values are positive real numbers. Therefore, we used Gaussian distribution as output.
>
> We would like to clarify that while output for a simple pass is parameters of a Gaussian, to capture the complex forecast distribution, we sample from individual Gaussian distributions output by the model via Monte-Carlo sampling by performing inference multiple times.
>
>
> ## References
>
> [1] Wu, Dongxia, et al. "DeepGLEAM: a hybrid mechanistic and deep learning model for COVID-19 forecasting." arXiv
>
> [2] Reich, Nicholas G., et al. "A collaborative multiyear, multimodel assessment of seasonal influenza forecasting in the United States." PNAS 2019
>
> [3] Adhikari, Bijaya, et al. "Epideep: Exploiting embeddings for epidemic forecasting." KDD 2019
>
> [4] UMass-Amherst Influenza Forecasting Center of Excellence. COVID-19 Forecast Hub; 2020. Accessible online at https://github.com/reichlab/covid19-forecast-hub.
>
> [5] Volodymyr Kuleshov, Nathan Fenner, and Stefano Ermon. Accurate uncertainties for deep learning using calibrated regression. ICML 2018
>
> [6] Jin, Xiaoyong, Yu-Xiang Wang, and Xifeng Yan. "Inter-Series Attention Model for COVID-19 Forecasting." SDM 2021
>
> [7] Qian, Zhaozhi, Ahmed M. Alaa, and Mihaela van der Schaar. "When and How to Lift the Lockdown? Global COVID-19 Scenario Analysis and Policy Assessment using Compartmental Gaussian Processes." NeurIPS 2020
>
> [8] Allen-Zhu, Zeyuan, Yuanzhi Li, and Yingyu Liang. "Learning and generalization in overparameterized neural networks, going beyond two layers." NeurIPS 2019
>
> [9] Arora, Sanjeev, et al. "Fine-grained analysis of optimization and generalization for overparameterized two-layer neural networks." ICML 2019.
>
> [10] Arora, Sanjeev, Nadav Cohen, and Elad Hazan. "On the optimization of deep networks: Implicit acceleration by overparameterization." ICML 2018.
>
> [11] Arora, Sanjeev, et al. "Stronger generalization bounds for deep nets via a compression approach." ICML 2018.
>
> [12] Du, Simon, et al. "Gradient descent finds global minima of deep neural networks." ICML 2019.

---

> > ### Comment · Reviewer_mtym · 2021-08-26
> > **willing to increase my score a little**
> >
> > Thank you for clarifying some issues. I still feel the method is very complex and I don't really trust it to work for other kinds of epi forecasting problems. However,  based on the evidence you have presented, it seems to work for the particular case of flu forecasting (for this US dataset) . Since this seems to outperform prior work on the same problem, I think it is possibly worth publishing as a poster.

---

### Official Review · Reviewer_ENNP · 2021-07-15

**Rating:** 5
**Confidence:** 4

**Summary:**

This paper a functional neural process model called EPIFNP for accurate and truth-worthy epidemic forecasting. EPIFNP is empirically shown with fast speed, accurate results and explainability.

**Limitations And Societal Impact:**

YES

**Main Review:**

I am not sure whether EPIFNP is specifically designed for epidemic forecasting. If so, what is the main difference between epi-forecasting and others and what is your main point towards this difference. If not, what if EPIFNP tested on other datasets, such as demand and weather forecasting?

What is the computation efficiency? EPIFNP uses variational inference and a correlation graph, which seems computational expensive.

For the baselines, they are all specially designed for epidemic forecasting. Back to the aforementioned problem, if EPIFNP is not designed for epidemic forecasting, some more general baseline should be included here, such as: calibration methods, post-hoc methods using isotonic regression, deep ensemble. I have seen some of them are included in your baselines, but they looks specially designed for epidemic forecasting. What is their differences?
* Alex Kendall and Yarin Gal. What uncertainties do we need in Bayesian deep learning for computer vision? Advances in neural information processing systems, pages 5574–5584, 2017.
* BalajiLakshminarayanan,AlexanderPritzel,andCharlesBlundell.Simpleandscalablepredic- tive uncertainty estimation using deep ensembles. Advances in neural information processing systems, pages 6402–6413, 2017.
* Volodymyr Kuleshov, Nathan Fenner, and Stefano Ermon. Accurate uncertainties for deep learning using calibrated regression. International Conference on Machine Learning, pages 2796–2804, 2018.
* Peng Cui, Wenbo Hu, Jun Zhu. Calibrated Reliable Regression using Maximum Mean Discrepancy. Advances in neural information processing systems, 2020.
* Song H, Diethe T, Kull M, et al. Distribution calibration for regression[C]//International Conference on Machine Learning. PMLR, 2019: 5897-5906.

**Time Spent Reviewing:**

2

---

> ### Author Response · Authors · 2021-08-10
> **Response to Reviewer 1**
>
> We thank the reviewer for their valuable comments.
>
> **I am not sure whether EPIFNP is specifically designed for epidemic forecasting. If so, what is the main difference between epi-forecasting and others and what is your main point towards this difference...**
>
>
> Our goal was to focus on the specific application of accurate and *well-calibrated* epidemic forecasting. Therefore, we specifically went deep into this application, proposed a unifying framework to provide accurate, well-calibrated and interpretable predictions, and compared against appropriate state-of-art baselines in this research area. This task also has multiple unique features for which we designed our model. Non-trivial continuously evolving season patterns [1] were leveraged via the correlation graph on latent representations. The need for reliable uncertainty measures for forecasts was solved by introducing the probabilistic GP-based neural process framework.
>
> We also provide important case studies showcasing EpiFNP's adaptability to uncertainty in our specific domain by studying the anomalous H1N1 and Covid-19 seasons where other top models failed. This showcases the robustness and reliability of our method for epidemic forecasting.
>
> However, our approach can easily be extended to other forecasting domains like climate and retail as mentioned by the reviewer. Our framework merges the expressive power of deep sequential models with the flexibility and interpretability of non-parametric Gaussian processes.
>
>
> **What is the computation efficiency? EPIFNP uses variational inference and a correlation graph, which seems computationally expensive.**
>
> Our model actually takes less training time than non-trivial deep learning baselines: EpiDeep, RNP and BNN. The training curve of our method also converges faster than these deep learning baselines.
>  Furthermore, using ensembles over EpiDeep, SARIMA and GRU for probabilistic forecasts multiply their computational cost. We provide the training time of the non-trivial baselines and EpiFNP as below:
>
> | Model                 | EpiFNP | SARIMA | EB | DD | BNN | RNP | GP | EpiDeep |
> |-----------------------|--------|--------|----|----|-----|-----|----|---------|
> | Training Time (min)  | 20     | 7      | 6  | 6  | 25  | 26  | 6  | 35      |
>
> Our model is time-efficient due to two reasons: (1) We use the
> reparameterization trick [4] for variational inference, which enables fast model
> training via standard backpropagation and stochastic gradient descent.
> Variational inference is thus not time-consuming in our method. (2) We
> design the correlation graph as a sparse binary graph, which actually improves
> computation efficiency instead of hurting it. Specifically, the correlation
> graph samples a small subset of reference points to directly leverage pattern
> similarity, which can speed up training by virtue of sparsity. In
> contrast, RNP uses an attention mechanism that is more computation- and
> memory-intensive.
>
>
>
>
>
> **For the baselines, they are all specially designed for epidemic forecasting. Some more general baseline should be included here, such as: calibration methods, post-hoc methods using isotonic regression, deep ensemble**
>
>
> We did include the well-known general baselines. As described in *Lines 272-290*, we introduce *two groups of baselines*, one related to flu forecasting literature and the other related to *general ML uncertainty methods*. Some flu forecasting methods can only provide point estimate prediction and thus we use **ensembles** to quantify their predictive uncertainty. The general ML uncertainty methods include **Monte Carlo Dropout, Bayesian Neural Networks and Recurrent Neural processes**. They are not specially designed for epidemic forecasting.
>
> We did not include the post-hoc calibration methods due to two reasons: (1) They
> require additional validation data (ideally from the same distribution with test
> data), but such validation data are unavailable in practical epidemic
> forecasting scenarios. (2) The benefit of performing post-hoc calibration is
> very limited if a model's probabilistic forecasts are already well-calibrated.
> As these post-hoc calibration methods are mentioned by the review, we still
> tested the effects of two well-known post-hoc calibration methods [2,3] on
> EpiFNP and the four best performing baselines. We observe that EpiFNP doesn't
> benefit much from post-hoc calibration methods due to its already
> well-calibrated forecasts. However, they improve the calibration scores of other
> baselines (sometimes at the cost of prediction accuracy). But *EpiFNP is still
> clearly the best performing model*. We can add these results in the final
> version.
>
>
>
> |         |          | RMSE |      |      | MAPE  |       |       | LS   |      |      | CS    |       |       |
> |---------|----------|------|------|------|-------|-------|-------|------|------|------|-------|-------|-------|
> | Model   | Post-Hoc | K=2  | K=3  | K=4  | K=2   | K=3   | K=4   | K=2  | K=3  | K=4  | K=2   | K=3   | K=4   |
> | EpiFNP  | None     | 0.48 | 0.79 | 0.78 | 0.089 | 0.128 | 0.123 | 0.56 | 0.84 | 0.89 | 0.068 | 0.081 | 0.035 |
> |         | Iso      | 0.49 | 0.81 | 0.79 | 0.09  | 0.124 | 0.119 | 0.56 | 0.86 | 0.9  | 0.08  | 0.09  | 0.07  |
> |         | DC       | 0.44 | 0.74 | 0.77 | 0.088 | 0.114 | 0.117 | 0.55 | 0.75 | 0.86 | 0.07  | 0.08  | 0.035 |
> | RNP     | None     | 0.61 | 0.98 | 1.18 | 0.13  | 0.22  | 0.29  | 3.34 | 3.61 | 3.89 | 0.43  | 0.38  | 0.34  |
> |         | Iso      | 1.77 | 2.26 | 2.18 | 0.18  | 0.27  | 0.28  | 2.55 | 2.62 | 3.12 | 0.18  | 0.23  | 0.24  |
> |         | DC       | 1.73 | 2.17 | 2.25 | 0.18  | 0.27  | 0.31  | 1.53 | 1.84 | 2.05 | 0.13  | 0.12  | 0.15  |
> | GP      | None     | 1.28 | 1.36 | 1.45 | 0.21  | 0.22  | 0.26  | 2.02 | 2.12 | 2.27 | 0.24  | 0.25  | 0.28  |
> |         | Iso      | 2.24 | 2.51 | 2.72 | 0.34  | 0.34  | 0.38  | 1.97 | 2.13 | 2.16 | 0.094 | 0.12  | 0.11  |
> |         | DC       | 2.15 | 2.68 | 2.72 | 0.32  | 0.37  | 0.39  | 1.94 | 2.07 | 2.04 | 0.09  | 0.11  | 0.1   |
> | EpiDeep | None     | 0.73 | 1.13 | 1.81 | 0.14  | 0.23  | 0.33  | 4.26 | 6.37 | 8.75 | 0.24  | 0.15  | 0.42  |
> |         | Iso      | 1.02 | 1.25 | 1.94 | 0.16  | 0.24  | 0.34  | 2.46 | 4.58 | 4.64 | 0.21  | 0.11  | 0.19  |
> |         | DC       | 1.15 | 1.28 | 1.74 | 0.17  | 0.26  | 0.32  | 2.11 | 3.97 | 3.65 | 0.18  | 0.14  | 0.21  |
> | MCDP    | None     | 2.24 | 2.41 | 2.61 | 0.46  | 0.51  | 0.6   | 9.62 | 10   | 10   | 0.24  | 0.32  | 0.34  |
> |         | Iso      | 2.36 | 2.58 | 2.53 | 0.45  | 0.47  | 0.59  | 6.72 | 9.64 | 10   | 0.14  | 0.26  | 0.31  |
> |         | DC       | 2.31 | 2.44 | 2.52 | 0.44  | 0.48  | 0.57  | 6.31 | 8.24 | 10   | 0.15  | 0.22  | 0.25  |
>
> Table 1: Effect of post-hoc calibration on point estimate and calibration scores. Iso and DC are post-hoc methods in [2] and [3] respectively.
>
>
>
> ## References
>
> [1] Adhikari, Bijaya, et al. "Epideep: Exploiting embeddings for epidemic forecasting." KDD 2019.
>
> [2] Volodymyr Kuleshov, Nathan Fenner, and Stefano Ermon. Accurate uncertainties for deep learning using calibrated regression. ICML 2018
>
> [3] Song H, Diethe T, Kull M, et al. Distribution calibration for regression ICML 2019
>
> [4] Kingma, Diederik P., and Max Welling. "Auto-encoding variational bayes." ICML 2014

---

### Decision · Program_Chairs · 2021-09-27

**Decision:**

Accept (Poster)

**Comment:**

There was significant interest in this paper, as the topic of epidemic forecasting is indeed one that has the attention of the community at the current time.  Reviewers had mixed opinions about this paper, but overall found that the most important concerns raised were indeed addressed by the author responses.

In particular, reviewer mtym's concern that the model may be prone to overfitting was reasonably addressed by the empirical evaluation.  (And indeed, this would not be the first model in ML literature to show that over-parameterization does not necessarily lead to overfitting.)

Reviewer EENP's questions around whether the model is specifically designed for this application area were also well addressed in rebuttal, along with useful additional detail around computational efficiency and the potential impact of post-hoc calibration.

Overall, I can see no reason not accept this paper as a poster at this point, with the clear expectation of course that the authors will incorporate the feedback from reviewers into revising their paper, and include the additional results from the responses into the final paper.